# Navigating the Pandemic: Shifts in Breast Reconstruction Trends and Surgical Decision-Making in the United States

**DOI:** 10.3390/jcm13144168

**Published:** 2024-07-16

**Authors:** Seung Eun Hong, Daihun Kang

**Affiliations:** 1Department of Plastic and Reconstructive Surgery, School of Medicine, Ewha Womans University, Seoul 07804, Republic of Korea; monkeyhong@hanmail.net; 2Department of Plastic and Reconstructive Surgery, Ewha University Seoul Hospital, Seoul 07804, Republic of Korea

**Keywords:** breast reconstruction, COVID-19, implant, autologous, immediate, delayed, DIEP, DTI

## Abstract

**Background**: This study aimed to investigate the impact of the COVID-19 pandemic on breast reconstruction trends in the United States, focusing on implant-based and autologous techniques, as well as the timing of reconstruction (immediate vs. delayed). **Methods**: A retrospective analysis of data from 2015 to 2022 from the American Society of Plastic Surgeons’ National Plastic Surgery Statistics was conducted. Annual trends in breast reconstruction procedures were analyzed, comparing the pre-pandemic (2015–2019) and pandemic (2020–2022) periods. A sensitivity analysis was performed to assess the impact of missing data from 2019 and 2021. **Results**: The total number of breast reconstructions increased from 106,338 in 2015 to 151,641 in 2022. The proportion of implant-based reconstructions decreased from 81.41% pre-pandemic to 76.51% during the pandemic (*p* < 0.001), with a notable rise in direct-to-implant procedures from 10.37% to 19.12% (*p* < 0.001). Autologous reconstructions increased from 18.59% to 23.49% (*p* < 0.001). Among autologous techniques, DIEP flaps remained the most popular. Immediate reconstruction rates increased from 72.61% pre-pandemic to 75.57% during the pandemic, while delayed reconstruction rates decreased from 27.39% to 24.43% (*p* < 0.001). The sensitivity analysis confirmed the robustness of these trends. **Conclusions**: During the COVID-19 pandemic, there was a significant shift towards autologous breast reconstruction techniques, particularly DIEP flaps, and an increase in immediate reconstructions in the United States. Within implant-based reconstructions, the proportion of DTI procedures increased while the proportion of tissue expander procedures decreased. These changes likely reflect adaptations in surgical practice and decision-making processes in response to the unique challenges posed by the pandemic, rather than alterations in surgical techniques themselves. The insights gained from this study can help guide plastic surgeons and healthcare providers in preparing for future pandemics or similar disruptions.

## 1. Introduction

Breast cancer is the most prevalent cancer among women worldwide, with an estimated 2.3 million new cases diagnosed in 2020 [1]. Mastectomy, the surgical removal of the entire breast, remains a common treatment option for women with breast cancer. Breast reconstruction, the restoration of the breast form and function following mastectomy, plays a crucial role in the physical and psychological well-being of patients [2].

The choice between implant-based and autologous reconstruction techniques depends on various factors, including patient preferences, oncologic considerations, and surgeon expertise [3]. Implant-based reconstruction utilizes silicone or saline implants to recreate the breast mound, while autologous reconstruction employs the patient’s own tissue, such as abdominal tissue or back muscle, to form the breast [4]. In recent years, there has been a growing trend towards implant-based reconstruction, with studies reporting a steady increase in their utilization [5]. This trend may be attributed to several factors, including advances in implant technology, improved surgical techniques, and patient preferences for shorter operative times and faster recovery [6].

To date, there have been numerous studies on trends in breast reconstruction and their changes over time [5,7,8]. However, there is a notable lack of research focusing on the changes that occurred during the COVID-19 pandemic. The authors sought to investigate the potential surgical changes brought about by this unprecedented global health crisis, particularly its impact on patients requiring breast reconstruction.

The recent pandemic caused by SARS-Coronavirus-2 has profoundly affected healthcare systems worldwide, and it is essential to understand its impact on specific medical practices such as breast reconstruction. Moreover, considering the history of similar viral outbreaks, such as SARS-CoV-1, and the potential emergence of future viruses like SARS-CoV-3, it is crucial to review past records and changes. This proactive analysis and preparation can help predict and respond to changes that may occur during future pandemics. These findings are particularly relevant for healthcare policymakers and medical practitioners who must allocate resources and design protocols that can withstand similar disruptions in the future.

This study aims to address this knowledge gap by analyzing the American Society of Plastic Surgeons’ (ASPS) data on breast reconstruction procedures from 2015 to 2022, comparing the pre-pandemic and pandemic periods. The authors hypothesize that the COVID-19 pandemic has led to an overall decrease in breast reconstructions due to the need to minimize hospital visits and reduce recovery times. Additionally, we anticipate a preference for delayed over immediate reconstructions and for implant-based over autologous reconstructions, with a particular increase in DTI procedures over tissue expander methods.

## 2. Materials and Methods

### 2.1. Data Source and Study Design

This retrospective study utilized data from the American Society of Plastic Surgeons’ (ASPS, Arlington Heights, IL, USA) National Plastic Surgery Statistics, spanning from 2015 to 2022. However, due to disruptions in data collection and reporting caused by the strain on healthcare systems during the COVID-19 pandemic, complete data for the years 2019 and 2021 were not available. Despite the missing data, the available information provides significant insights into trends and shifts in breast reconstruction practices during the pandemic.

The dataset is freely accessible on the ASPS statistics website. It includes procedural data reported by ASPS member surgeons and other physicians certified by boards recognized by the American Board of Medical Specialties. The comprehensive dataset is anonymized and provided for public access, ensuring compliance with ethical standards for data privacy and confidentiality.

### 2.2. Ethical Considerations

Given the public accessibility and fully anonymized nature of the ASPS data, this study did not require Institutional Review Board (IRB) approval. The data were used strictly within the bounds of ethical guidelines for academic research. By utilizing publicly available data that adhere to stringent privacy standards, this study maintains the highest ethical standards, ensuring that all analyses are performed without compromising individual privacy.

### 2.3. Variables and Definitions

Breast reconstruction procedures were categorized as implant-based (tissue expander/implant and direct-to-implant) or autologous (transverse rectus abdominis myocutaneous [TRAM], deep inferior epigastric perforator [DIEP], latissimus dorsi, and other flaps). Reconstruction timing was classified as immediate (performed at the same time as mastectomy) or delayed (performed after mastectomy).

The study period was divided into pre-pandemic (2015–2019) and pandemic (2020–2022) periods, with 2020 considered the start of the pandemic period, aligning with the World Health Organization’s declaration of the COVID-19 outbreak as a global pandemic in March 2020.

### 2.4. Statistical Analysis

Descriptive statistics were calculated to summarize the trends in the number and proportion of implant-based and autologous reconstructions, as well as immediate and delayed procedures. The Shapiro–Wilk test was employed to assess the normality of the data distribution. As the data violated the normality assumption, non-parametric tests were utilized for comparisons between the pre-pandemic (2015–2019) and pandemic (2020–2022) periods. The relative proportions of reconstruction methods (implant-based vs. autologous) and timing (immediate vs. delayed) were compared using chi-square tests. Statistical significance was set at a two-sided *p*-value < 0.05.

To address the potential impact of missing data from 2019 and 2021, a sensitivity analysis was performed. Three scenarios were considered for imputing the missing values: (1) using the average values of 2018 and 2020, (2) using the average values of 2020 and 2022, and (3) using linear interpolation between 2018 and 2022. Chi-square tests were conducted for each scenario to compare the proportions of reconstruction methods and timing between the pre-pandemic and pandemic periods. All statistical analyses were conducted using SPSS version 29.0.2.0 (IBM Corporation, Armonk, NY, USA).

## 3. Results

### 3.1. Overall Trends in Breast Reconstruction

A comprehensive analysis of breast reconstruction procedures in the United States from 2015 to 2022 was conducted using data from the ASPS Database (Table 1). The findings revealed a significant increase in the annual number of breast reconstructions performed, with a 42.5% rise from 106,338 in 2015 to 151,641 in 2022, highlighting the growing demand for these procedures (Figure 1). 

To better understand the impact of the COVID-19 pandemic on breast reconstruction trends, the data were divided into pre-pandemic (2015–2019) and pandemic (2020–2022) periods (Table 2). While detailed data on reconstruction type and timing were unavailable for 2019 and 2021, the overall trends could still be observed.

During the pre-pandemic period, implant-based reconstructions accounted for 81.41% of all procedures, with autologous reconstructions making up the remaining 18.59%. However, during the pandemic, the proportion of implant-based reconstructions decreased to 76.51%, while that of autologous reconstructions increased to 23.49% (*p* < 0.01).

Similarly, the timing of reconstructions also exhibited a change during the pandemic period. Immediate reconstructions increased from 72.61% in the pre-pandemic period to 75.57% (*p* < 0.01), while delayed reconstructions decreased from 27.39% to 24.43% (*p* < 0.01).

### 3.2. Implant-Based vs. Autologous Reconstruction

During the COVID-19 pandemic, a notable shift was observed in the adoption of breast reconstruction methods. The proportion of implant-based reconstructions decreased from 81.41% in the pre-pandemic period (2015–2019) to 76.51% during the pandemic (2020–2022) (*p* < 0.01) (Table 2). Conversely, autologous reconstructions significantly increased from 18.59% to 23.49% (*p* < 0.001). This shift suggests a growing preference for autologous tissue reconstruction during the pandemic.

Within the implant-based reconstruction category, a significant shift towards direct-to-implant (DTI) procedures was observed. The proportion of DTI procedures increased from 10.37% to 19.12% during the pandemic (*p* < 0.001), while tissue expander procedures decreased from 71.05% to 57.38% (*p* < 0.001) (Table 2).

Among the autologous techniques, DIEP flaps remained the most popular throughout the study period, accounting for 41.60% of all flap-based reconstructions in 2015 and 58.95% in 2022 (Table 3). The proportion of TRAM flaps decreased from 25.63% in 2015 to 9.68% in 2022, while latissimus dorsi flaps decreased from 29.20% to 15.28% during the same period (Figure 2). Notably, the proportion of other flaps, which include less common techniques such as superior gluteal artery perforator (SGAP) and transverse upper gracilis (TUG) flaps, increased from 3.57% in 2015 to 16.09% in 2022 (Table 3). This increase in other flaps may reflect a growing interest in alternative autologous options, possibly due to advancements in microsurgical techniques and a desire for more customized reconstructions.

### 3.3. Immediate vs. Delayed Reconstruction

The timing of reconstructions also shifted, with immediate reconstructions increasing from 72.61% in the pre-pandemic period to 75.57% during the pandemic, and delayed reconstructions decreasing from 27.39% to 24.43% (Table 2). These changes were statistically significant (*p* < 0.001), as confirmed by chi-square tests.

A more detailed analysis of the annual trends in reconstruction timing is presented in Table 4. The proportion of immediate reconstructions showed a steady increase from 70.50% in 2015 to 76.68% in 2020, with a slight decrease to 74.57% in 2022. Conversely, delayed reconstructions exhibited a gradual decline from 29.50% in 2015 to 23.32% in 2020, followed by a small increase to 25.43% in 2022. Figure 3 graphically emphasizes these shifts, highlighting the notable rise in immediate reconstructions starting from 2018.

The upward trend in immediate reconstructions suggests a strategic clinical pivot towards performing reconstructions at the same time of mastectomy. This shift may reflect a response to the pandemic’s operational constraints, such as the need to minimize hospital visits and optimize resource utilization. Additionally, the benefits of immediate reconstruction, including improved esthetic outcomes and patient satisfaction, may have contributed to this trend.

### 3.4. Trends in Implant-Based Breast Reconstruction Techniques

In the realm of implant-based reconstructions, the data denote a paradigm shift toward single-stage procedures. The incidence of DTI approaches saw a statistically significant increase from 10.37% pre-pandemic to 19.12% during the pandemic (*p* < 0.001), juxtaposed by a notable decrease in tissue expander/implant procedures from 71.05% to 57.38% (*p* < 0.001). These trends are quantitatively depicted in Figure 4 and delineated in Table 2.

The comprehensive annual data outlined in Table 5 further illustrate the evolution of DTI and tissue expander/implant case volumes. The proportion of DTI procedures steadily increased from 10.22% in 2015 to 29.98% in 2022, while tissue expander procedures decreased from 89.78% to 70.02% during the same period. Notably, the most prominent shift towards DTI occurred between 2020 and 2022, coinciding with the pandemic period. The total number of implant-based reconstructions also exhibited a consistent growth trend, increasing from 86,013 in 2015 to 117,957 in 2022, despite the challenges posed by the pandemic.

These findings suggest a growing preference for single-stage implant-based reconstruction, particularly DTI, which may be attributed to factors such as advancements in surgical techniques, the use of acellular dermal matrices, and the desire to minimize hospital visits and exposure during the pandemic. The simultaneous decrease in tissue expander procedures further underscores this shift in clinical practice.

### 3.5. Sensitivity Analysis

A sensitivity analysis was conducted to evaluate the potential impact of missing data from 2019 and 2021 on the observed trends in breast reconstruction. Three scenarios were considered for imputing the missing values by (1) using the average values of 2018 and 2020, (2) using the average values of 2020 and 2022, and (3) using linear interpolation between 2018 and 2022.

In scenario 1, the imputed values for implant-based and autologous reconstructions in 2019 were 93,351 (77.18%) and 27,382 (22.82%), respectively. For 2021, the imputed values were 110,721 (76.75%) and 33,504 (23.25%). In scenario 2, the imputed values for 2019 were 93,351 (77.18%) and 27,382 (22.82%), while for 2021, they were 110,721 (76.75%) and 33,504 (23.25%). In scenario 3, the imputed values for 2019 were 93,351 (77.18%) and 27,382 (22.82%), and for 2021, they were 110,721 (76.75%) and 33,504 (23.25%).

The chi-square test was performed for each scenario to compare the proportions of implant-based and autologous reconstructions between the pre-pandemic and pandemic periods. In all three scenarios, the results remained statistically significant (*p* < 0.001), with a decrease in implant-based reconstructions and an increase in autologous reconstructions during the pandemic period.

Similarly, for immediate and delayed reconstructions, the imputed values in scenario 1 were 90,409 (75.62%) and 29,323 (24.38%) for 2019, and 109,370 (75.82%) and 34,855 (24.18%) for 2021. In scenario 2, the imputed values for 2019 were 90,409 (75.62%) and 29,323 (24.38%), while for 2021, they were 109,370 (75.82%) and 34,855 (24.18%). In scenario 3, the imputed values for 2019 were 90,409 (75.62%) and 29,323 (24.38%), and for 2021, they were 109,370 (75.82%) and 34,855 (24.18%).

The chi-square test results for immediate and delayed reconstructions remained statistically significant (*p* < 0.001) in all three scenarios, with an increase in immediate reconstructions and a decrease in delayed reconstructions during the pandemic period.

The consistency of the results across the three imputation scenarios suggests that the missing data for 2019 and 2021 did not significantly influence the observed trends in implant-based vs. autologous reconstructions and immediate vs. delayed reconstructions.

## 4. Discussion

The COVID-19 pandemic has profoundly impacted healthcare delivery across various specialties, necessitating adaptations in surgical practices and decision-making processes [9]. Our study provides a comprehensive analysis of the pandemic’s influence on breast reconstruction trends in the United States, revealing significant shifts in reconstruction techniques and timing. The most notable changes include a decreased proportion of implant-based reconstructions, increased adoption of autologous techniques, particularly DIEP flaps, and a higher rate of immediate reconstructions. These findings align with anecdotal reports from plastic surgeons during the pandemic [10] and highlight the potential impact of pandemic-related factors on breast reconstruction practices [11].

### 4.1. Implant-Based Reconstruction: Shift towards Direct-to-Implant Techniques

Although the overall proportion of implant-based reconstructions decreased during the pandemic, a significant shift towards DTI techniques was observed within this category. The proportion of DTI procedures nearly doubled, increasing from 10.37% to 19.12% (*p* < 0.001), while tissue-expander-based reconstructions decreased from 71.05% to 57.38% (*p* < 0.001) (Table 2). This trend may be attributed to several factors, including the desire to minimize hospital visits, reduce postoperative complications, and optimize resource utilization during the pandemic [12].

DTI techniques, particularly prepectoral implant placement with the use of acellular dermal matrices (ADMs), have gained popularity in recent years [13]. Prepectoral implant placement involves positioning the implant above the pectoralis major muscle, as opposed to the traditional subpectoral placement. ADMs are used to provide additional soft tissue coverage and support, reducing the risk of implant-related complications such as capsular contracture and malposition [14]. The increased adoption of DTI techniques with ADMs during the pandemic may reflect a preference for a single-stage procedure that minimizes the need for multiple surgeries and hospital visits.

However, it is essential to consider the potential long-term implications of the increased use of DTI techniques. While studies have demonstrated comparable outcomes and patient satisfaction between DTI and two-stage expander/implant reconstructions [15], long-term data on the safety and effectiveness of prepectoral implant placement with ADMs are still limited [16].

### 4.2. Rise in Autologous Reconstruction: Emphasis on DIEP Flaps

Our study revealed a significant increase in the proportion of autologous reconstructions during the pandemic, with DIEP flaps being the most popular technique. The proportion of DIEP flaps among all autologous reconstructions increased from 41.60% in 2015 to 58.95% in 2022 (Table 3). This shift towards autologous reconstruction, particularly DIEP flaps, is noteworthy considering the general trend towards less complex surgeries during the pandemic [10].

Despite the challenges posed by the COVID-19 pandemic, such as the need to minimize hospital visits and the preference for simpler surgical procedures [10], autologous reconstruction, especially DIEP flaps, demonstrated a remarkable increase. This suggests that the benefits of autologous reconstruction, including better long-term results and patient satisfaction [17], outweighed the perceived advantages of simpler implant-based procedures.

Several factors may have contributed to this trend. Firstly, advancements in microsurgical techniques have made DIEP flaps a more reliable and efficient procedure, with improved outcomes and shorter recovery times [8,18]. Secondly, the increasing availability of information on the long-term benefits of autologous reconstruction has helped patients make more informed decisions, leading to a higher demand for DIEP flaps [19,20].

Moreover, the emphasis on patient-centered care and shared decision-making has likely played a role in the increased adoption of DIEP flaps [21]. As patients become more involved in the decision-making process and prioritize long-term results and natural-looking outcomes, the preference for autologous reconstruction, particularly DIEP flaps, may grow.

It is important to note that while the DIEP flap technique itself did not undergo significant changes during the pandemic [22], the observed trends reflect shifts in patient preferences and hospital practices. These shifts were likely driven by a combination of factors, including advancements in surgical techniques, increased patient awareness, and a focus on patient-centered care, rather than changes in the surgical technique itself.

### 4.3. Shift towards Immediate Reconstruction

The COVID-19 pandemic has also influenced the timing of breast reconstruction, with a notable shift towards immediate reconstruction. The proportion of immediate reconstructions increased from 72.61% in the pre-pandemic period to 75.57% during the pandemic (*p* < 0.01), while delayed reconstructions decreased from 27.39% to 24.43% (*p* < 0.01) (Table 2). This shift may be attributed to several factors, including the desire to minimize hospital visits, reduce the overall treatment timeline, and alleviate patient anxiety [10,11,12,23].

Immediate reconstruction offers several advantages in the context of the pandemic. By combining mastectomy and reconstruction into a single procedure, patients can reduce the number of hospital visits and minimize their exposure to potential COVID-19 transmission [24]. Additionally, immediate reconstruction allows patients to wake up from the mastectomy with a reconstructed breast, which may provide psychological benefits and improve quality of life [18].

However, immediate reconstruction may not be suitable for all patients, particularly those with advanced disease or those requiring post-mastectomy radiation therapy [8]. The decision to pursue immediate reconstruction should be made through a shared decision-making process, considering the individual patient’s oncologic status, treatment plan, and personal preferences.

### 4.4. Implications for Clinical Practice and Patient Care

The findings of this study have important implications for clinical practice and patient care in the era of COVID-19. Plastic surgeons should be aware of the shifting trends in breast reconstruction and adapt their practices accordingly. The increased preference for autologous reconstruction, particularly DIEP flaps, may require surgeons to receive additional training or collaborate with microsurgery specialists to meet the growing demand.

Furthermore, the shift towards immediate reconstruction highlights the need for close coordination between the breast cancer treatment team and the reconstructive surgeons. Effective communication and planning are essential to ensure that patients receive the optimal timing and type of reconstruction based on their individual needs and preferences.

Patient education and shared decision-making are also crucial considering these changing trends. Patients should be provided with comprehensive information about the various reconstruction options, including the benefits and risks of each technique, and the potential impact of the pandemic on their surgical journey. Surgeons should engage in open and honest discussions with patients to help them make informed decisions that align with their values and goals.

Moreover, the increased adoption of DTI techniques with ADMs underscores the importance of long-term follow-up and surveillance. Plastic surgeons should establish appropriate post-operative monitoring protocols to identify and manage any implant-related complications promptly. Patients should be educated about the signs and symptoms of potential complications and encouraged to maintain regular follow-up appointments.

### 4.5. Future Research Directions

The findings of this study provide a foundation for several interesting avenues of future research. Firstly, the long-term outcomes and patient satisfaction associated with the increased use of DTI techniques during the pandemic should be investigated. Prospective studies with extended follow-up periods could evaluate the safety and effectiveness of various DTI approaches, including the use of innovative materials and techniques such as mesh or robotic-assisted surgery. As DTI methods continue to evolve, comparing the outcomes of different surgical approaches and materials will be crucial for guiding evidence-based decision-making and optimizing patient care.

Secondly, the impact of the pandemic on autologous reconstruction, particularly DIEP flaps, warrants further exploration. Future studies could assess the outcomes, complication rates, and patient satisfaction associated with DIEP flaps performed during the pandemic, while also investigating the potential influence of advancements in microsurgical techniques and robotic-assisted surgery on surgical outcomes and efficiency.

Moreover, it would be interesting to examine the potential growth in customized, microsurgery-based reconstruction in the absence of the COVID-19 pandemic’s disruptive effects. By controlling for the pandemic’s impact, researchers could estimate the true demand for these advanced techniques and predict their future adoption rates. This information would be valuable for healthcare providers, policymakers, and patients alike, as it could guide resource allocation, training programs, and patient education initiatives.

Thirdly, the psychosocial impact of the shift towards immediate reconstruction during the pandemic should be examined. Qualitative studies exploring patient experiences, quality of life, and satisfaction with immediate reconstruction in the context of the pandemic could provide valuable insights. Investigating the effectiveness of innovative support interventions, such as online support groups and telemedicine consultations, in promoting patients’ mental well-being and quality of life would also be worthwhile.

Lastly, future research should address the limitations of this study, particularly the missing data for certain years (2019 and 2021), and aim to validate the findings using prospective designs and comprehensive data collection. Researchers should explore strategies to mitigate the impact of missing data, such as imputation methods or sensitivity analyses, to ensure the robustness of the results. Additionally, potential confounding factors, such as changes in patient demographics, cancer stage distribution, and adjuvant therapy trends, should be considered when interpreting the observed shifts in breast reconstruction trends during the pandemic.

## 5. Conclusions

The COVID-19 pandemic has significantly influenced breast reconstruction trends in the United States, with a shift towards autologous techniques, particularly DIEP flaps, and an increase in immediate reconstructions. These changes reflect adaptations in surgical practice and decision-making processes in response to the unique challenges posed by the pandemic.

Contrary to the initial hypothesis, the total number of breast reconstructions performed during the pandemic increased. While there was a rise in DTI procedures as anticipated, the hypothesized decrease in autologous reconstructions and overall breast reconstructions was not observed. Instead, there was a significant increase in autologous reconstructions, particularly DIEP flaps, and a decrease in the proportion of implant-based reconstructions, although DTI procedures within this category increased. These unexpected findings suggest that factors such as advancements in microsurgical techniques and the dedication of healthcare providers to maintain high standards of care may have counterbalanced the anticipated effects of the pandemic on breast reconstruction trends.

Our analysis indicates that while the DIEP flap technique itself did not undergo significant changes during the pandemic [22], the observed trends reflect shifts in patient preferences and hospital practices. These shifts were likely driven by the need to minimize hospital visits and adapt to pandemic-related constraints, rather than changes in the surgical technique itself.

The insights gained from this study provide valuable lessons for the plastic surgery community in preparing for potential future viral outbreaks, such as SARS-CoV-3, as mentioned in the introduction. By understanding the impact of the COVID-19 pandemic on breast reconstruction trends, surgeons and healthcare providers can develop proactive strategies to ensure the continuity of high-quality care during similar disruptions in the future.

As we navigate the ongoing impact of COVID-19, plastic surgeons must remain vigilant, adapt their practices, and prioritize patient safety and outcomes. The findings of this study underscore the importance of patient education, shared decision-making, and long-term follow-up in the context of changing breast reconstruction trends. By understanding the implications of these trends, we can optimize patient care, inform clinical practice guidelines, and ensure that breast cancer patients receive the highest quality of care in the face of the challenges posed by the COVID-19 pandemic and any future global health crises.

## Figures and Tables

**Figure 1 jcm-13-04168-f001:**
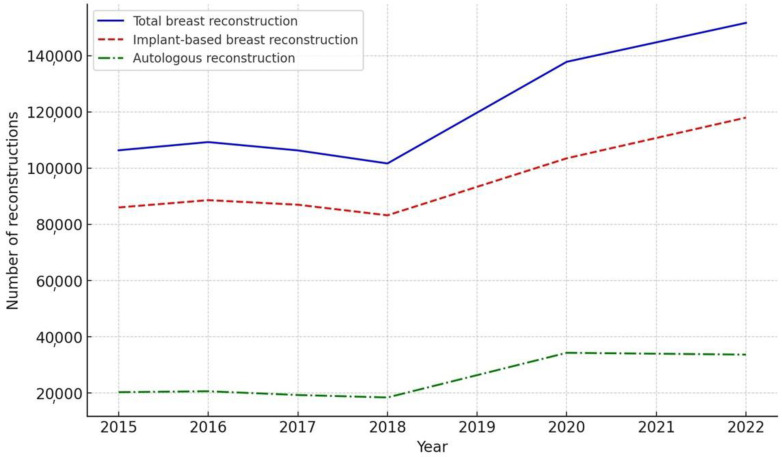
Trends in breast reconstruction procedures in the United States from 2015 to 2022. While the total number of breast reconstructions has notably increased since 2018, this growth has been predominantly due to a significant rise in implant-based reconstructions.

**Figure 2 jcm-13-04168-f002:**
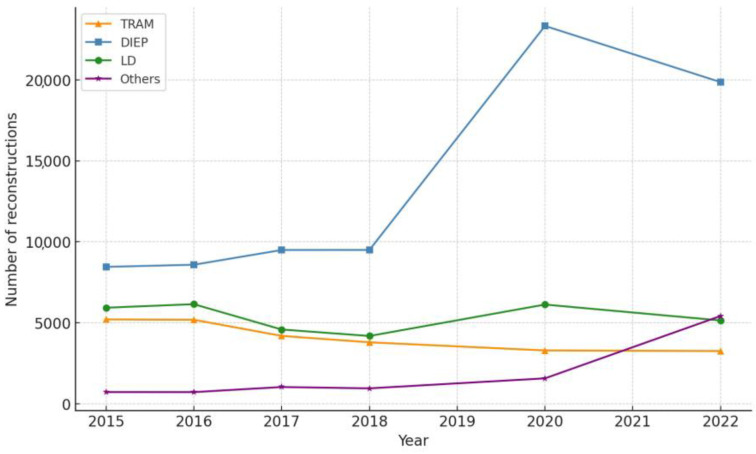
Trends in autologous breast reconstruction techniques in the United States from 2015 to 2022. This figure delineates the number of autologous breast reconstructions performed, categorized by technique: transverse rectus abdominis myocutaneous (TRAM) flap, deep inferior epigastric perforator (DIEP) flap, latissimus dorsi (LD) flap, and other techniques, such as superior gluteal artery perforator (SGAP) and transverse upper gracilis (TUG) flaps. Throughout the study period, the DIEP flap notably became the most favored technique for autologous reconstruction. The data also reveal a decline in TRAM flap reconstructions and a mark rise in the ‘Others’ category, surpassing the frequency of LD flaps, particularly during the pandemic years.

**Figure 3 jcm-13-04168-f003:**
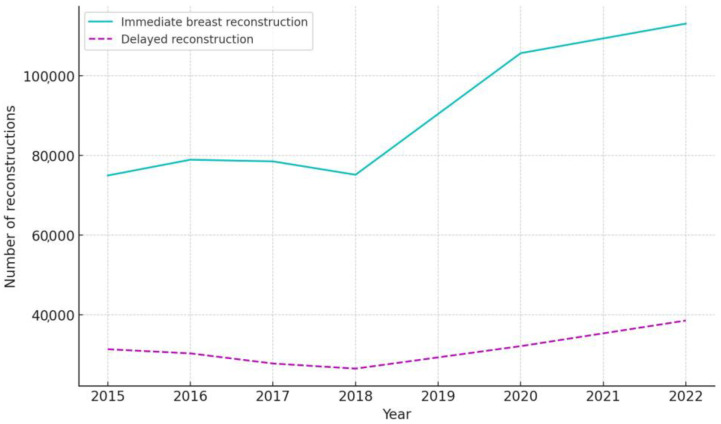
Comparison of immediate and delayed breast reconstruction trends from 2015 to 2022. The number of immediate reconstructions increased steadily over the years, while delayed reconstructions showed a slight but gradual increase. However, the gap between immediate and delayed reconstructions widened during the pandemic period (2020–2022), indicating a more pronounced increase in immediate reconstructions compared to delayed reconstructions.

**Figure 4 jcm-13-04168-f004:**
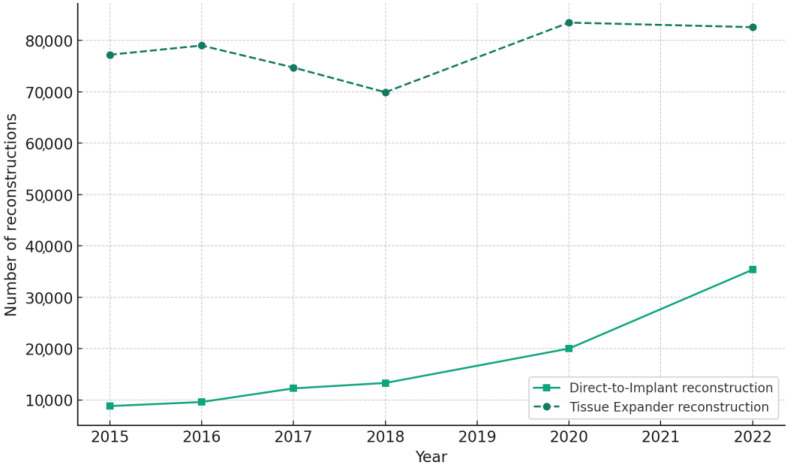
Trends in implant-based breast reconstruction techniques (direct-to-implant and tissue expander) from 2015 to 2022. The number of direct-to-implant reconstructions increased significantly over the years, particularly during the pandemic period (2020–2022). Tissue expander reconstructions also increased but at a slower rate compared to direct-to-implant procedures.

**Table 1 jcm-13-04168-t001:** Breast reconstruction procedures in the US: 2015–2022.

Year	Total Breast Reconstruction	Implant-Based Reconstructions	Autologous Reconstructions	Immediate Reconstructions	Delayed Reconstructions
2015	106,338	86,013 (80.77%)	20,325 (19.23%)	74,968 (70.50%)	31,370 (29.50%)
2016	109,256	88,606 (81.07%)	20,650 (18.93%)	78,932 (72.25%)	30,324 (27.75%)
2017	106,295	86,979 (81.96%)	19,316 (18.04%)	78,511 (73.86%)	27,784 (26.14%)
2018	101,657	83,216 (81.86%)	18,441 (18.14%)	75,153 (74.93%)	26,504 (26.07%)
2019	135,996	N/A	N/A	N/A	N/A
2020	137,808	103,485 (75.50%)	34,323 (24.50%)	105,665 (76.68%)	32,143 (23.32%)
2021	N/A	N/A	N/A	N/A	N/A
2022	151,641	117,957 (78.00%)	33,684 (22.00%)	113,075 (74.57%)	38,566 (25.43%)

Data for 2019 and 2021 were not available (N/A).

**Table 2 jcm-13-04168-t002:** Comparison between pre-pandemic and pandemic breast reconstruction in the US.

Reconstruction Type/Timing	Pre-Pandemic (*n*, %)	Pandemic (*n*, %)	*p*-Value
Implant-based	344,814 (81.41%)	221,442 (76.51%)	<0.001
DTI	43,922 (10.37%)	55,358 (19.12%)	
Tissue Expander/implant	300,892 (71.05%)	166,084 (57.38%)	
Autologous	78,732 (18.59%)	68,007 (23.49%)	
TRAM	18,393 (4.34%)	6556 (2.26%)	
DIEP	36,032 (8.51%)	43,181 (14.92%)	
LD	20,862 (4.93%)	11,275 (3.89%)	
Others	3445 (0.81%)	6995 (2.42%)	
Immediate	307,564 (72.61%)	218,740 (75.57%)	
Delayed	115,982 (27.39%)	70,709 (24.43%)	
Total	423,546	289,449	

**Table 3 jcm-13-04168-t003:** Trends in autologous breast reconstruction techniques in the US: 2015–2022.

Year	Total Autologous Procedures	DIEP Flaps	TRAM Flaps	Latissimus Dorsi Flaps	Other Flaps
2015	20,325	8455 (41.60%)	5210 (25.63%)	5934 (29.20%)	726 (3.57%)
2016	20,650	8585 (41.57%)	5190 (25.13%)	6151 (29.79%)	724 (3.51%)
2017	19,316	9495 (49.16%)	4194 (21.71%)	4589 (23.76%)	1038 (5.37%)
2018	18,441	9497 (51.50%)	3799 (20.60%)	4188 (22.71%)	957 (5.19%)
2020	34,323	23,324 (67.95%)	3297 (9.61%)	6128 (17.85%)	1574 (4.59%)
2022	33,684	19,857 (58.95%)	3259 (9.68%)	5147 (15.28%)	5421 (16.09%)

Note: DIEP, deep inferior epigastric perforator; TRAM, transverse rectus abdominis myocutaneous. Data presented as n (%). Data for 2019 and 2021 were not available.

**Table 4 jcm-13-04168-t004:** Comparative analysis of immediate and delayed breast reconstruction rates.

Year	Total	Immediate Reconstructions	Delayed Reconstructions	Immediate (%)	Delayed (%)
2015	106,338	74,968	31,370	70.50%	29.50%
2016	109,256	78,932	30,324	72.25%	27.75%
2017	106,295	78,511	27,784	73.86%	26.14%
2018	101,657	75,153	26,504	73.93%	26.07%
2020	137,808	105,665	32,143	76.68%	23.32%
2022	151,641	113,075	38,566	74.57%	25.43%

Note: Data for 2019 and 2021 were not available.

**Table 5 jcm-13-04168-t005:** Yearly breakdown of implant-based reconstruction procedures.

Year	Total IBR	DTI	Tissue Expander	DTI (%)	Tissue Expander (%)
2015	86,013	8794	77,219	10.22%	89.78%
2016	88,606	9587	79,019	10.82%	89.18%
2017	86,979	12,246	74,733	14.08%	85.92%
2018	83,216	13,295	69,921	15.98%	84.02%
2020	103,485	19,998	83,487	19.32%	80.68%
2022	117,957	35,360	82,597	29.98%	70.02%

Note: IBR, implant-based breast reconstruction; DTI, direct-to-implant. Data for 2019 and 2021 were not available.

## Data Availability

The original data presented in the study are openly available in https://www.plasticsurgery.org/news/plastic-surgery-statistics (accessed on 14 May 2024).

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
