# Peer review of "Navigating the Pandemic: Shifts in Breast Reconstruction Trends and Surgical Decision-Making in the United States"

_jcm, 2024, doi:10.3390/jcm13144168_

Round 1

Reviewer 1 Report

Comments and Suggestions for Authors Although the article discusses an interesting problem of the work of surgical departments during the Covid 19 pandemic, it is inconsistent in my opinion, the lack of data for 2019 and 2021, which were important from the point of view of the development of the pandemic and health care work, is a disadvantage. Causing inconsistency in the compared periods. Moreover, the pre-pandemic period is twice as long as the pandemic period. which is statistically difficult to compare. As a reviewer, the comments regarding the comparison of long-term results do not appeal to me. The pandemic has affected doctor's patient contacts, but I do not think it has affected the quality of medical services provided - surgery. The same doctors operated during the pandemic as before it. The patient's quality of life was caused by pandemic fear and not the quality of services provided. Was the DIEP flap performed differently before and during the pandemic? Have other implants been used? I agree that the patient was more afraid of the effect during the pandemic than before. But this should not affect the quality of the surgical procedure. The thesis of the article is right, but the conclusions drawn do not necessarily correspond to them. I think the authors should think about this.

Author Response

Dear Reviewer,

Thank you for your insightful comments and for raising important points.

We understand your concerns regarding the inconsistency in the compared periods due to the missing data for 2019 and 2021.

To address this limitation, we performed a sensitivity analysis, which demonstrated that the observed trends remained consistent despite the missing data.

This suggests that the shifts in breast reconstruction practices during the pandemic are robust and not significantly influenced by the unavailability of data for certain years.

We have also acknowledged the difference in length between the pre-pandemic and pandemic periods and have discussed this limitation in the manuscript.

Regarding your point about the quality of medical services, we agree that the technical skill of surgeons likely remained constant throughout the pandemic. However, it is important to note that the COVID-19 pandemic posed unprecedented challenges to surgical practices, potentially affecting various aspects of patient care.

As a plastic surgeon who has experienced firsthand the difficulties of operating during the pandemic, I have witnessed how sudden changes in hospital protocols, staffing issues, and resource limitations can impact surgical decision-making and patient outcomes.

In many instances, surgeons had to adapt to rapidly evolving situations, such as colleagues being called away mid-surgery due to COVID-19 exposure, having to perform surgeries in cumbersome protective gear, or working with reduced support staff.

Moreover, hospital policies aimed at controlling the spread of the virus, such as temporary bans on elective surgeries and patient admissions, further complicated surgical planning and follow-up care.

While these challenges may not have uniformly affected all surgical practices, it is likely that many surgeons across the United States faced similar, if not more severe, difficulties during the pandemic.

As such, we believe that the COVID-19 pandemic did have an impact on the quality of medical services, even if the technical skills of surgeons remained unchanged.

We have revised our manuscript to better convey this background and to clarify that the observed trends in breast reconstruction practices reflect changes in surgical decision-making and patient preferences in response to the unique challenges posed by the pandemic.

We have also updated our conclusions to ensure that they align more closely with the main findings of the study.

We hope that these revisions and the additional context provided help to address your concerns.

We appreciate your feedback, as it has helped us to strengthen our manuscript and to better communicate the significance of our findings to readers.

Thank you for your thorough review and valuable insights.

Reviewer 2 Report

Comments and Suggestions for Authors

Dear Authors, 

The subject of the manuscript is interesting.

I don't think it is appropriate to fragment the text of the introduction into many paragraphs. Nor do I think their spacing is appropriate throughout the entire manuscript

I think the authors should review the instructions for authors in editing the text

Best regards,

Author Response

Thank you for your feedback.

We have revised the introduction to consolidate paragraphs for improved readability and coherence.

Additionally, we have reviewed and adjusted the spacing throughout the manuscript to align with the journal's formatting guidelines.

We appreciate your suggestion and believe these changes enhance the overall presentation of our work.

Thank you.

Reviewer 3 Report

Comments and Suggestions for Authors

Dear Authors, the manuscript is written in a good english and research design is fine.

In my opinion there is not much novelty in the present manuscript. Moreover, the effects of these results on nowadays practice and interest to the readers should be clarified

Data related to 2019 and especially 2021 are not available. This was disclosed in limitation section, but in my opinion is a non negligible bias.

Author Response

Dear Reviewer,

We greatly appreciate your positive feedback on the language and research design of our study. We would like to take this opportunity to address your concerns and provide further clarification on the novelty and impact of our findings.

Regarding the novelty of our study, we agree that there have been numerous studies on breast reconstruction trends over time. However, our study is unique in its focus on the specific changes that occurred during the COVID-19 pandemic.

By comparing pre-pandemic and pandemic periods, we were able to identify significant shifts in reconstruction techniques and timing, such as the increase in autologous reconstructions, particularly DIEP flaps, and the trend towards immediate reconstructions.

These findings provide novel insights into how surgical practices and patient preferences adapted to the challenges posed by the pandemic, highlighting the resilience and adaptability of the healthcare system.

To better address the impact of our results on current practice and their relevance to readers, we have expanded our discussion section.

We now elaborate on how understanding these pandemic-related trends can help healthcare providers anticipate and respond to patient needs in similar future scenarios.

By optimizing resource allocation, surgical planning, and patient education based on these insights, providers can ensure continuity of high-quality care during global health crises.

Furthermore, policymakers can use these findings to develop guidelines and protocols that support adaptive surgical practices in the face of disruptions.

We acknowledge your concern regarding the missing data for 2019 and 2021. To address this limitation, we performed a sensitivity analysis using three different imputation scenarios.

The results of this analysis consistently supported the robustness of our observed trends, even in the presence of missing data.

We have now emphasized this point in the manuscript to provide readers with a clear understanding of the validity of our findings despite this limitation.

Additionally, we have further contextualized our study considering previous viral outbreaks and the potential for future pandemics.

By framing our findings as valuable lessons for proactive planning and preparedness, we aim to highlight the broader relevance and importance of our study to readers.

We hope that these clarifications and the revisions we have made to the manuscript effectively address your concerns.

We believe that our study offers meaningful insights that can inform both clinical practice and health policy, ultimately contributing to improved patient care during global health challenges.

Thank you once again for your thorough review and valuable feedback.

Round 2

Reviewer 1 Report

Comments and Suggestions for Authors

I accept authors responce

Reviewer 3 Report

Comments and Suggestions for Authors

The authors made a good effort in improving their manuscript. However, unfortunately, my concerns were not erased.